# Microstructural Analysis and Mechanical Properties of TiMo_20_Zr_7_Ta_15_Si_x_ Alloys as Biomaterials

**DOI:** 10.3390/ma13214808

**Published:** 2020-10-28

**Authors:** Adriana Savin, Mihail Liviu Craus, Alina Bruma, František Novy, Sylvie Malo, Milan Chlada, Rozina Steigmann, Petrica Vizureanu, Christelle Harnois, Vitalii Turchenko, Zdenek Prevorovsky

**Affiliations:** 1Nondestructive Testing Department, National Institute for Research and Development for Technical Physics, 700050 Iasi, Romania; steigmann@phys-iasi.ro; 2Frank Laboratory for Neutron Physics, Joint Institute for Nuclear Research, Dubna 141980, Russia; turchenko@jinr.ru; 3National Institute of Standards and Technology, Gaithersburg, MD 20899, USA; bruma.alina@outlook.com; 4Department of Materials Engineering, University of Zilina, 010 26 Zilina, Slovak Republic; frantisek.novy@fstroj.uniza.sk; 5Normandie Université, ENSICAEN, UNICAEN, CNRS, CRISMAT, 14000 Caen, France; sylvie.malo@ensicaen.fr (S.M.); christelle.harnois@ensicaen.fr (C.H.); 6Institute of Thermomechanics, Academy of Sciences of the Czech Republic, 182 00 Prague, Czech Republic; chlada@it.cas.cz (M.C.); zp@it.cas.cz (Z.P.); 7Faculty of Materials Science and Engineering, Technical University Gheorghe Asachi, 700050 Iasi, Romania; peviz2000@yahoo.com

**Keywords:** titanium alloys, microstructure, mechanical properties, resonant ultrasound spectroscopy, acoustic emission, inhomogeneities

## Abstract

TiMoZrTaSi alloys appertain to a new generation of metallic biomaterials, labeled high-entropy alloys, that assure both biocompatibility as well as improved mechanical properties required by further medical applications. This paper presents the use of nondestructive evaluation techniques for new type of alloys, TiMo_20_Zr_7_Ta_15_Si_x_, with x = 0; 0.5; 0.75; 1.0, which were obtained by vacuum melting. In Ti alloys, the addition of Mo improves tensile creep strength, Si improves both the creep and oxidation properties, Zr leads to an α crystalline structure, which increases the mechanical strength and assures a good electrochemical behavior, and Ta is a β stabilizer sustaining the formation of solid β-phases and contributes to tensile strength improvement and Young modulus decreasing. The effects of Si content on the mechanical properties of the studied alloys and the effect of the addition of Ta and Zr under the presence of Si on the evolution of crystallographic structure was studied. The influence of composition on fracture behavior and strength was evaluated using X-ray diffraction, resonant ultrasound spectroscopy (RUS) analyses, SEM with energy dispersive X-ray spectroscopy, and acoustic emission (AE) within compression tests. The β-type TiMo_20_Zr_7_Ta_15_Si_x_ alloys had a good compression strength of over 800 MPa, lower Young modulus (69.11–89.03 GPa) and shear modulus (24.70–31.87 GPa), all offering advantages for use in medical applications.

## 1. Introduction

Implanted prosthetics are regulated by the FDA [1] as class III devices after demonstrating reasonable assurance of safety and efficacy. All metal devices, regardless of alloying elements, are subjected to a variety of non-clinical tests (bench and/or animal) followed by specific FDA documentation or national/international standards.

The clinical success of modern medical implants depends on the integration with the bone to resist functional loading [2]. Their mechanical properties depend on the manufacturing process or thermal and mechanical treatments, processes that can change the microstructure [3]. The stainless steels (iron-based alloys), Co and Ti alloys are very useful for medical implants. As an alternative to those metals, commercially pure titanium (cpTi) and its alloys were promoted due to their physical and mechanical features being of interest in prosthodontics or for endosseous implants, as well as plates and screws for bones fixtures. Recent research in material science and continuous technological developments have stimulated innovations in design and the diversification of alloys used in implant fabrication [4].

Used in physiological conditions, alloys are exposed to mechanical and biological factors that can affect long-term use. Some materials for implants contain metals that are normally found in small quantities in the human body, having an essential role in biological functioning (Cu, Mn, Co, Zn, Ca, etc.), having a critical role for structure, and/or proper functioning. Disruption of biological functions in the presence of implants can occur through associated signs or symptoms, when both essential and non-essential metals are present in concentrations which are either too high or too low. These biological events can be avoided by identifying and evaluating all factors that influence biocompatibility [5] according to ISO standards [6].

Titanium presents few crystallographic phases in biocompatible alloys due the high corrosion strength and adequate hardness/weight ratio [7].

In pure Ti, the modification of the crystalline β-phase cannot be obtained at ambient temperature, even by quenching with very high cooling rates; the β-phase passes into the α’-hexagonal form through a martensitic type of transformation. The difference between implant and bone stiffness can lead to the β-phase based implant weakening [8]. The contemporary and complex micro-alloyed systems research has been focused on reducing the rigidity and the effect of stress shielding. For titanium alloys, α, α + β, and β categories are known according to their equilibrium constitution, function of the type of alloying elements, and ratios [9].

Ti and Mo are compatible elements that, when combined, can form binary alloys leading to stable crystal structures, optimal for biomedical integration. Part of the elements used for titanium alloys ensure β stabilization, such as Mo (which improves the tensile strength of creep) [10] or Si (which improves the creep and oxidation properties). The β-phase proportion in Ti alloys affects the mechanical and corrosive properties [11,12], therefore, a compromise needs to be achieved between strength and plasticity, by alloying the TiMoSi ternary alloy with stabilizing elements for medical prostheses [13]. Previously published studies [14,15] focused on determining the properties of TiMo_y_Si_x_ alloys with x = 0; 0.5; 0.75; 1.0 and y = 15; 20; 25, and showed that Mo assures the stability of β-Ti (spatial group Im–3m) by lowering the transition temperature of α-Ti to β-Ti to ambient temperature [16], and ensuring a fine structure of titanium alloy. TiMoZrTaSi takes part from the new generation of metallic biomaterials labeled high-entropy alloys (HEAs) [17], that assure both biocompatibility as well as improved mechanical properties required by further medical applications. Zr and Ta act as consolidation elements: the alloying with Zr leads to an α crystalline structure which increases the mechanical strength and assures a good electrochemical behavior; Ta is a β stabilizer sustaining the formation of solid β-phases in the titanium alloy. The addition of Ta contributes to the improving of tensile strength and the decrease in the Young modulus.

The studied samples are from TiMo_20_Si_x_ alloys with x = 0; 0.5; 0.75; 1.0 and doped with 7 wt.% Zr and 15 wt.% Ta. Selection of alloying elements was made according to [18], referring to bond order (B_o_) values and metal d-orbital energy level (M_d_) on the phase stability diagram.

Combining characterization techniques such as energy-dispersive spectroscopy (EDS) and X-ray diffraction (XRD) analysis as well as nondestructive evaluation (NDE) with ultrasound and resonant ultrasound spectroscopy (RUS), the physical properties of these compounds were evaluated. After that, acoustic emission (AE) was carried out during the compression testing.

This paper presents few nondestructive testing methods for selected biocompatible alloys TiMo_y_Si_x_ when Mo percent is 20 wt.% and Si varies from 0 to 1.0 wt.% in the presence of Ta and Zr doping. The mechanical properties were noninvasively evaluated. The studies have been completed with the analysis of crystal structure of samples and microstructural characterization based on their dependence by microstructure, phase composition and the alloying elements content, knowing that the evolution of Ti alloy microstructures during the production has a special importance [8].

## 2. Materials and Methods

### 2.1. Samples Preparation

In [15], the elastic modulus of TiMo_15_Zr_7_Ta_5_ and TiMo_15_Si_0.5_ are compared. It has shown that, for a lower elastic modulus comparable with the human bone, a combination of ratios of alloying elements in TiMoZrTa is required, so that the value should reach that of cortical bone.

Using raw materials with high purity as starting materials, mini-ingots were prepared, with the composition empirically designed as to reduce elastic modulus and increase the compatibility of TiMo_20_Zr_7_Ta_15_Si_x_ with elastic properties of the cortical bone. The procedure of alloying was developed by the Faculty of Materials Science and Engineering, Gh. Asachi Technical University, Iasi, Romania [16]. In [19], a modeling approach is presented for understanding the hardening mechanisms of Ti alloys, seeking to evaluate the contribution of the effect of solute elements to the hardening mechanism. For binary and ternary β metastable titanium alloys (Ti-Mo; Ti-Mo-Si), the neutral alloying additions as Ta and Zr assure the formation of microstructures in β-phase and improve tensile strength while decreasing the Young modulus [20], and improving the strength at fatigue crack propagation and deformation. Doping with Zr and Ta has a minor influence on the β-transition temperature, but together with Si qualitatively influences the stabilization of the β-phase. Si has low solubility in both the α- and β-phases, contributing to hardening and decreased ductility, but can improve creep resistance at high temperatures, increase corrosion resistance and increase fluidity during casting. The stabilizing effect strongly depends on the wt.% of the other alloying elements in the compound, Ta being known to cause solid solution hardening [21], however the experimental results cannot be discernible due to the transition phases and changes in the deformation mechanisms [22].

The mini-ingot quaternary TiMoZrTa alloys investigated were obtained from raw precursor elements, melted in a vacuum arc remelting VAR MRF ABJ 900 furnace (Allenstown, NH, USA) in high purity Ar atmosphere and a vacuum (4 × 10^−3^ mbar); which assured temperature control. The remelting was carried out with minimal overheating to reduce the segregation [23]. The mini-ingots were obtained by melting (at temperatures higher than 3000 °C), then remelted between 2 to 7 times to assure their homogeneity.

### 2.2. Surface Analysis; Microstructure

The samples were fixed in EpoFix Resin (Struers, Rotherham, UK) and their surface was polished and etched to reveal the microstructure. The microstructure was observed on sections, after mechanical processing, by optical microscopy using AxioCam MRc5-Zeiss (Jena, Germany) and by electron microscopy using SEM JEOL JSM7200F (JEOL Ltd., Tokyo, Japan), equipped with a Bruker EDS analyzer (BRUKER AXS Inc., Madison, WI, USA).

After that, the samples were re-polished and the Vickers micro hardness HV0.5 was measured. Few methods for material characterization (i.e., phase identification via X-ray diffraction (XRD) and optical microscopy) were employed. Tests to identify the phase composition and the type of crystal structure of crystallographic phase were carried out by XRD—Xpert Pro MPD PANalytical diffractometer (Phillips, New York, NY, USA) with CoKα radiation, and a Bruker AXS D8 Advance diffractometer (Karlsruhe, Germany) with Cu-Kα radiation, respectively [16].

### 2.3. Ultrasound Measurements

The longitudinal (C_l_) and transverse (C_t_) ultrasound speeds of waves through the samples were determined using the method described in [14] using two sets of wide-band transducers: for the longitudinal wave sensor, G5 KB GE [14]; and for the transversal wave sensor, MB4Y GE. The PR 5077 pulse/receiver system was used for emission and reception of the signals. The timing of flight and digitizing of the signal were carried out with digital oscilloscope Wave Runner 64Xi, LeCroy (LeCroy Corporation, Chestnut Ridge, NY, USA). Elastic properties (E, G and ν) were calculated from the ultrasound (US) time of flight.

### 2.4. Mechanical Properties

#### 2.4.1. Resonant Ultrasound Spectroscopy (RUS)

RUS was employed to evaluate the quality of the studied samples. RUS implies the exploration of the resonant structure of a compact sample on the basis of resonance frequency modification according to ASTM E2001-18 [24], for the detection and assessment of variations and mechanical properties of a test object. RUS techniques can employ small samples, where the evaluation is made from the entire volume; the method can also be applied for phase transition studies [25]. The resonance frequencies depend on mechanical properties (size, shape, density) and on the sample’s elasticity. Most resonance frequencies of a cube do not respond to analytical solutions, even if it is isotropic. However, there are analytical solutions offered by Mindlin-Lamé [26] and Damarest, whose formulations offer the possibility to manage anisotropy. For small samples of the order of centimeters, RUS is applicable for allowing the connection of low frequency stress strain methods and ultrasound time delay. Each frequency corresponds to a propagating wave beam, the maximum of resonance modes having 1/f period. (f is the resonance frequency). RUS is frequently combined with Finite Element Analysis (FEA) to identify the resonance frequency, *f_r_*. Typically, in conventional RUS, the sample is fixed between the sensors by “edge-to edge” contact [27], assuring a weak coupling with the sensors. An emission transducer (US Em) and a receiver transducer (US Re) type P111.O.06P3.1 with high bandwidth were used to carry out the experiment. The experimental assembly included an Agilent 4395A Network/Spectrum/Analyzer generating a frequency sweep in 1 kHz steps between 120–200 kHz, as shown in Figure 1.

#### 2.4.2. Acoustic Emission (AE)

AE is a highly accurate method for detecting active microscopic events in samples and for detecting the initiation/propagation of a crack [14]. In order to test the mechanical performances of TiMo_20_Si_x_Zr_7_Ta_15_ alloys designed and prepared in the form of specimens with different chemical composition, AE measurements were carried out during compression loading. Structural modifications in titanium alloys during compression loading were accompanied by different frequencies noises; AE analysis method showed information useful for the identification of failed mechanisms.

The samples were tested by AE during compressive tests on an INSTRON 1195 with 0.2 mm/min displacement speed. The compression tests were carried out up to an 80 kN load (the limit of the equipment was 100 kN). Crack and microcrack initiation and propagation during compressive deformation generated AE events, with the elastic waves propagating through the sample towards the piezoelectric transducers and converting to electric signals. These signals were amplified and distinguished after root-mean-square (RMS) calculations, the number of thresholds was counted (*N_c_*), count rate (*C_x_* = *dN_c_*/*dt*), intensity parameters, frequency spectrum, signal waveform, amplitude, etc., [27,28,29,30]. 

*C_x_* was evaluated as the number of AE signal crossings over the voltage level *x*, allowing the suppression of extraneous noise and the enhancement of higher energy processes. AE activity depends on damage and transformation processes during the compression of samples.
(1)$$RMS={\left(\frac{1}{T}\int_{0}^{T}a^{2}\left(t\right)dt\right)}^{\frac{1}{2}}$$

The RMS value is proportional to the emitted acoustic energy, which mostly characterizes running fracture processes such as micro-crack jumps in the sample.

The strength of materials can be predicted by recording the AE events from the initiation of microcracks [15,31,32,33]. The AE sensors (two DAEL IDK09) could not be applied directly to the sample, due its small dimensions, thus, they were glued onto support plates, acting as waveguides (Figure 2). 

Signals from the sensors were amplified by 20 and 40 dB, respectively, in two preamplifiers. Continuous AE signals recording with 14-bit amplitude resolution, and 2.5 MHz sampling by USB oscilloscope were employed. 

Standard microindentation hardness (HV) was measured according to ASTM E384 [34] with a ZwickRoell ZHµ Vickers micro-hardness tester with a loading of 500 gf/10 s, calculating the average of 6 measurements at 1 mm distance for each sample. 

## 3. Results and Discussions

### 3.1. Characterization of the Samples

#### 3.1.1. Mechanical Properties of Studied Alloys

The mechanical properties determined by the ultrasound method are presented in Table 1.

The results also include the binary alloy TiMo_20_ in order to emphasize the influence of doping with Ta and Zr on the density and Young modulus, as well as the influence of Si on the mechanical properties.

Table 1 shows that, compared to the binary/ternary alloy (sample #1 and #2), the presence of doping with 15 wt.% Ta (sample #3–#6) made the Young modulus decrease, thus the alloy was more elastic; a result that is in agreement with [35]. It is observed that the presence of Zr and Ta increased the density of the binary/ternary alloy, due to the large difference in density of Zr [36], while Si in both binary and ternary structures resulted in a decrease in density.

#### 3.1.2. Microstructure of Studied Alloys

XRD is used to investigate the surface layer up to 10 µm depth. Using XRD analysis, the crystal structures and lattice parameters of the constituent phases were determined.

Table 2 shows the Si concentration (x), average size of crystalline blocks (D), the value of microstrains (ε), lattice constants observed (a) and calculated (a_calc_), and unit cell volume (V) [16]. Here, B_o_ represents the covalent bond strength and M_d_ represents metal d-orbital energy level for TiMo_20_Zr_7_Ta_15_Si_x_ (x = 0.0; 0.5; 0.75; 1.0) alloy samples at room temperature. Table 3 presents the atomic concentration of alloying elements, and lattice constants and densities calculated using PowderCell and/or FullProf programs. The lattice constants were calculated from the number of atoms on the main diagonal of the elementary cell.

The main phase of the TiMoSi alloys is a Ti solid solution, being characterized by space group Im–3m. Increasing Si content leads to a decrease in the lattice constant when only one phase is present at the substitution of Ti with Si, Mo, Zr and Ta. Here, however, a maximum of the lattice constant was observed for x = 0.75. The calculated lattice value was obtained using the following equation:(2)$$a_{i}^{c}{}^{alc}=\frac{4}{\sqrt{3}}(C_{i}^{T}{}^{i}*1.40+C_{i}^{M}{}^{o}*1.45+C_{i}^{Z}{}^{r}*1.55+C_{i}^{T}{}^{a}*1.45+C_{i}^{S}{}^{i}*1.10)$$

The interpretation of the lattice constant values shown in Table 2, up to a Si concentration of 0.75 wt.% can be seen, as Si did not substitute with Ti, and probably formed some Si-based compounds. This supposition is justified by the presence of some small and broad maxima on the diffractograms. The calculated density from the experimental lattice constants was very close to the calculated density of the theoretical values of lattice constants (Table 3). We assumed that the weight of the unit cell was given by the nominal formula of the alloys. The design of titanium alloys and prediction of β stability and deformation mechanisms were suggested in [37] and is based on the calculations of electron structures. The electron parameters which predict the alloy stability are B_o_ and M_d_, in relation with the electric properties and metallic radius of atoms [19,38,39,40,41,42].

For an alloy, the average values ${\bar{B}}_{o}$ and ${\bar{M}}_{d}$ are given by compositional averages [41]
(3)$${\bar{B}}_{o}=\sum_{i=1}^{5}x_{i}B_{o}{}_{i}$$
and, respectively,
(4)$${\bar{M}}_{d}=\sum_{i=1}^{5}x_{i}M_{d}{}_{i}$$
where *x_i_* represents the atomic weight/atomic mass, and the *B_oi_*, *M_di_* values are of each *i*^th^ element in the alloy composition. Mo, Ta and Zr will contribute to the obtaining of high values for *B_oi_*, and therefore are added to the design of a good β-Ti alloy, with improved properties suitable for biomedical applications, such as reducing the elasticity modulus. According to [38], the values of the average *B_o_* and the average *M_d_* indicate that all casted alloys are in the stable β region. That is in agreement with the phase composition of the present alloys, which contain practically only the β-phase, with small concentrations of extrinsic phases (Figure 3).

It is possible that some maxima, due to some compounds with Si, cannot be identified due to the small dimensions of the mosaic blocks or very small concentrations.

The results emphasize that all the samples contained a low number of extrinsic phases; these phases are represented by two/one maxima, at the limit of the intensity errors. We considered a maximum when its intensity was at least three times larger than the intensity error.

Figure 4 presents the microstructure and semi-quantitative composition spectrum for the TiMo_20_Zr_7_Ta_15_ sample, showing a dendritic structure, irregular grain boundaries with several segregations, and large grains from dendritic colonies. The SEM and EDS analyses were performed at 15 kV, with a working distance of 9.9 mm.

The dendrite structures in Ti samples obtained by casting can be observed in Figure 5. A redistribution of the constituent elements during the solidification resulted in the formation of a dendritic structure.

A non-uniaxial fine dendrite structure of TiMo_20_Zr_7_Ta_15_Si_x_ samples is depicted in Figure 5. Due to the small dimensions of cast ingots, those dendrite structures were independent of the sample position and orientation. A tendency of evolution of the distance of the main dendrites axis dependent on Si content can be observed, but cannot be quantified using standard metallographic methods due to the random dendrite orientation.

It is also observed that, in addition to the structural body of the main dendrite with a white contrast, these structures appear with an inhomogeneous light gray contrast near the black regions visible in the Figure 4 EDS maps. These trends of microstructures can be explained by the tendency of Ti and Zr to be abundant in the inter-dendritic region [43], (as well as the tendency of Ta and Mo to be more abundant in dendrites).

### 3.2. RUS for Elasticity Measurements

RUS is used to determine the elastic properties of compact samples with regular geometries based on resonance frequencies [44,45]. According to [46] the elastic constants are connected to resonance frequency. In order to find resonant modes, a frequency sweep was made with a frequency generator in a range established for each sample by simulation. The fundamental mode’s frequency [46] is $f=\frac{m}{2h}\sqrt{\frac{G}{\rho }}$ with *h* as the side of the cube, *G* as the shear modulus, *ρ* as the density and *m* as an integer. To use this method, it is useful to determine the sample’s eigen frequencies and then to obtain an FEM model equivalent of eigen frequencies calculations, and finally, minimizing the object function (*F*) [47]:(5)$$F={\sum_{i}^{}w_{i}\left(f_{i}^{\left(p\right)}-f_{i}^{\left(m\right)}\right)}^{2}$$
where *f*^(*p*)^ and *f*^(*m*)^ are the computed and measured frequencies, respectively, and *w_i_* is the weight showing the reliability of the measurements. RUS is not entirely an experimental method, it presents an inverse problem [48], which most often implies a conjugate gradient method [49] minimizing the object function (*F*) using the determined frequencies. The symmetry produces more resonant modes at the same resonance frequency (*f_r_*). The presence of inhomogeneities can be detected by the modification of resonance spectra or by the variation of resonance frequencies (*f_r_*), and the splitting of the modes may be correlated with the dimensions of inhomogeneities [50].

Figure 6 presents the resonance spectra for samples #1; #3; #5 and #6 in the range from 120 to 220 kHz. 

The proposed method locates the fundamental frequency, knowing that the resonant modes for metals are sharp. Figure 6 shows that the samples had a different response in the amplitude and range of eigen frequencies, which can be correlated with the elastic properties of the samples, dimensions of grains and the crystalline structures (Table 2). The intervals of eigen frequencies present slight displacement (Table 2), and the resonant frequencies in the spectrum do not appear as separated peaks.

It can be observed that with the increase in Si content of over 0.75 wt.%, the frequency spectrum modifies its amplitude and shape and sharp peaks appear, which is specific to Si present in a Ti matrix, which is in good correlation with the values presented in Table 1. In addition, this denotes a modification of grain structure that seems to slightly decrease with the increase in Si content and leads to increasing hardness (Table 2). This modification of the structure, smaller than that of pure Ti, may be due to the hardening of the solid solution induced by the addition of Mo. The results show that the mechanical properties for the alloys were improved when the addition of Si was up to 0.75 wt.%.

The resonance frequencies were predicted by FEM simulation in SolidWorks 2019, a mesh of 28,264 nodes and 19,287 total elements. The typical response for the two modes obtained by simulation, extensional and flexural, for the studied samples #3 and #5 in a 120–250 kHz frequency range, was determined according the physical properties of samples [16].

From the RUS spectrum, it can be observed that, for sample #5, both extensional *f* = 189 kHz and flexural *f* = 151 kHz modes are shifted according to mechanical parameters from Table 1, as well as with the crystallographic structure from Table 2.

Figure 7 presents the elastic properties according to Table 1, for samples studied, showing that these values reached a minimum for Si content of 0.75 wt.%, as *ν* = 0.39 and *G* = 24.70 GPa. 

The results of RUS and US measurements can be in good concordance for homogeneous samples with high density and without voids and pores. Both methods assure the analysis in the volume of samples.

### 3.3. AE Response During Compressive Test

AE is used for monitoring the possible damage during compression tests, and assessing the influence of the microstructure over the final mechanical properties of titanium alloys. From data recorded during AE compression tests, the threshold crossing, RMS/s, and the level of stress and strain available at loading machine analog outputs are calculated [14].

To illustrate the spectral content of burst AE signals, an approach has been developed (described below), solving the problem of the extraction of AE hits, which can be relatively noisy and time-overlapping. As a simple tool, the thresholding of the signal envelope was performed (Figure 8). The continuous signal envelope was calculated as the convolution of the originally sampled signal with the Hamming window of an appropriately chosen length. AE hit impact was then defined as a signal part (light-green colored in Figure 8) corresponding to the multiple of time period *T*, in which the signal envelope was higher than detection level (set slightly above the noise). Multiplication coefficient, *k*, was set around the value *k* = 1.3 to capture the AE hit impact decay as well.

After the extraction of AE hits, the “correlation” method of spectral analysis was applied, considering the unsuitability of classical Fast Fourier Transform (FFT), namely the variable length of signals and resulting problems with common frequency resolutions. Analogically, compared to Fourier Transform, it is possible to compute the measure *p*_HD_ representing the inherence of frequency f_rel_ in signal *s*:(6)$$p_{HD}\left(f_{rel}\right)={\left|\sum_{n=0}^{N-1}s\left(n\right)e^{i\Pi nf_{rel}}\right|}^{2}$$
where arbitrary chosen frequencies *f_rel_* ∈ (0,1) are relative to the Nyquist frequency, Nq. All computations were made accordingly with the selected frequency resolution 1 kHz, while Nq = 500 kHz. Correlation values $p_{HD}$ for those 500 frequencies represented the estimate of a particular AE hit periodogram. To show the global changes of the spectral content of AE hits, partial periodograms of extracted hits detected within each second of the experiment were summed into corresponding column vectors of HD Power Spectral Distribution (PSD), and figured out as an alternative to classical spectrograms (Figure 8). By the above-mentioned method, only the parts of continuous signal with significant AE bursts were taken into account. In Figure 9, the spectral analysis of AE activities of Ti alloys for sample #5 and #6 are presented.

Performed compression tests of samples have shown a major influence of chemical composition on their fracture behavior and resulting strength due to different damage mechanisms. The formation of microcracks and their coalescence can be well documented by AE activity during the loading. Internal cracking manifested by the accompanying AE activity resulted in different fractures.

The compression curves have the characteristic S shape for sample #5 and sample #6. The AE signal (Figure 9) is very intense during the start of compression, close to the compressive failure point (sample #6); high amplitude AE events and a maximum at the yield point are presented. The AE activity is connected to the inflection points [51], and distinct differences in the compression behavior occur in relation to the Si content.

The AE signal in sample #6 can be attributed to the twining of the sample, which also has a higher modulus of elasticity. Tests have shown that the compressive strength of a Ti alloy is high enough (about 800 MPa), however internal cracking shown by the AE signal appeared at a stress of about 300 MPa, with rapid degradation to 600 MPa.

The decrease in AE activity in the region of plastic deformation of the sample could be due to the dynamics of structural modifications. After stopping the compression (maximum peaks of the stress curves) the material began to relax, producing only weak and apparently decreased secondary AEs.

The significantly higher AE signal in the case of alloys for which x = 1.0 compared to that for x = 0.75 can be explained by the differences between the internal microstructures. Most events take place around the possible dislocations where the crack initiates, forming microstructures, and their coalescence propagates towards the external surface in the compression direction, leading to high AE events during loading. The microcracks develop into the breaking of small particles, leading to the failure of the sample.

## 4. Conclusions

The changes of the mechanical properties and crystallographic structures of the TiMoZrTa quaternary alloy with variable doping by Si were investigated using various non-destructive evaluation (NDE) methods. The Si content in the TiMo_20_Zr_7_Ta_15_Si_x_ alloy was varied with x = [0.0; 0.5; 0.75; 1.0]. The main NDE investigation methods were RUS (resonant ultrasound spectroscopy), AE (acoustic emission) and X-ray diffraction. Those methods revealed the importance of the Ta and Zr content in the TiMo binary alloy, that together with NDE methods highlighted the influence of the Si addition on the mechanical properties and crystallographic structure. The changes in crystallographic structure showed that all the samples contained a small foreign phase in the error limit. The ascertained values of the average bond order (*B_o_*) and the averaged orbital energy level (*M_d_*) indicated that all casted alloys were in the stable β-region. RUS and compression tests showed an influence of the composition on the strength and fracture behaviors with various damage mechanisms.

The test results verified that TiMo_20_Zr_7_Ta_15_Si_x_ alloys have good mechanical behavior, with a Young modulus in the range of 69.11 to 89.03 GPa, and shear modulus from 24.70 to 31.87 GPa. All the tested samples were slightly affected by the Si content through their moduli with non-linear trends. The compressive strength of the studied Ti alloy, highlighted by AE, turned out to be over 800 MPa. Despite this, the internal cracking, which appeared in AE signals at 300 MPa stress, was followed by rapid failure at 600 MPa. The decrease in AE activity in the region of plastic deformation could be due to the structural modification dynamics. Although Si improves the creep strength of Ti alloys, AE test activity has shown that the plasticity is affected at a Si content higher than 0.75 wt.%.

The dependence of crystallographic structure on Si concentration is probably due to modification of the sample’s resonance spectrum symmetry, demonstrated by the splitting of the peaks at certain resonance frequencies in RUS.

Structural modifications of samples caused by different Si doping ratios could easily be monitored by RUS, emphasizing the changing of eigen frequencies. This led to the splitting of some spectral peaks, increasing their amplitudes and displacement of the RUS spectra from their initial position. The shape of the RUS spectrum and the resonance frequencies were also modified by the lower content of alloying elements and smaller elastic properties than those prescribed.

The analysis of the alloy microstructure dependence on the TiMo_20_Zr_7_Ta_15_Si_x_ components, as well as on the constitutive phases will be reported in detail in another paper.

## Figures and Tables

**Figure 1 materials-13-04808-f001:**
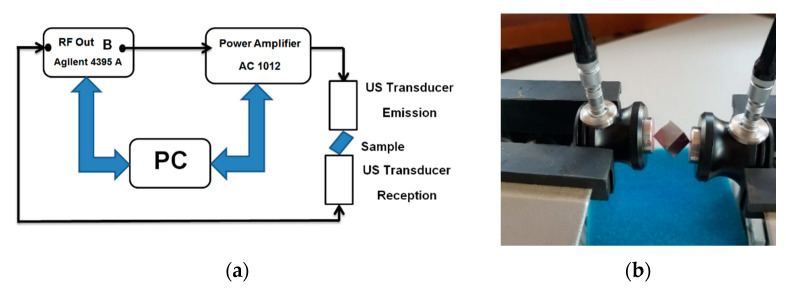
Resonant ultrasound spectroscopy (RUS) experimental set-up: (**a**) basic diagram; (**b**) detail of sample fixture.

**Figure 2 materials-13-04808-f002:**
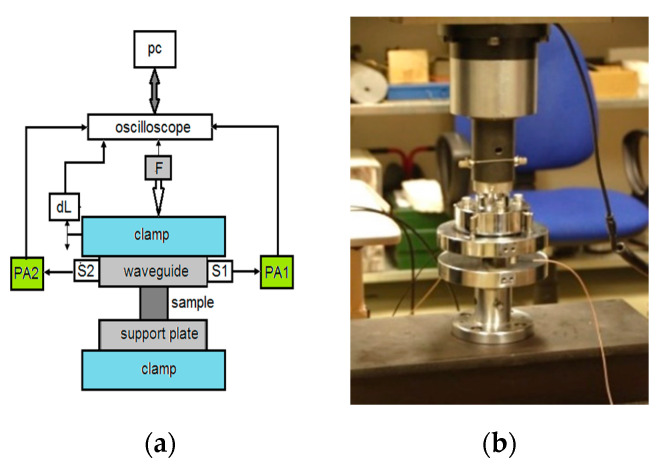
Experimental set-up of acoustic emission (AE): (**a**) basic diagram; (**b**) positioning of the sensors on the support plate ensuring a waveguide.

**Figure 3 materials-13-04808-f003:**
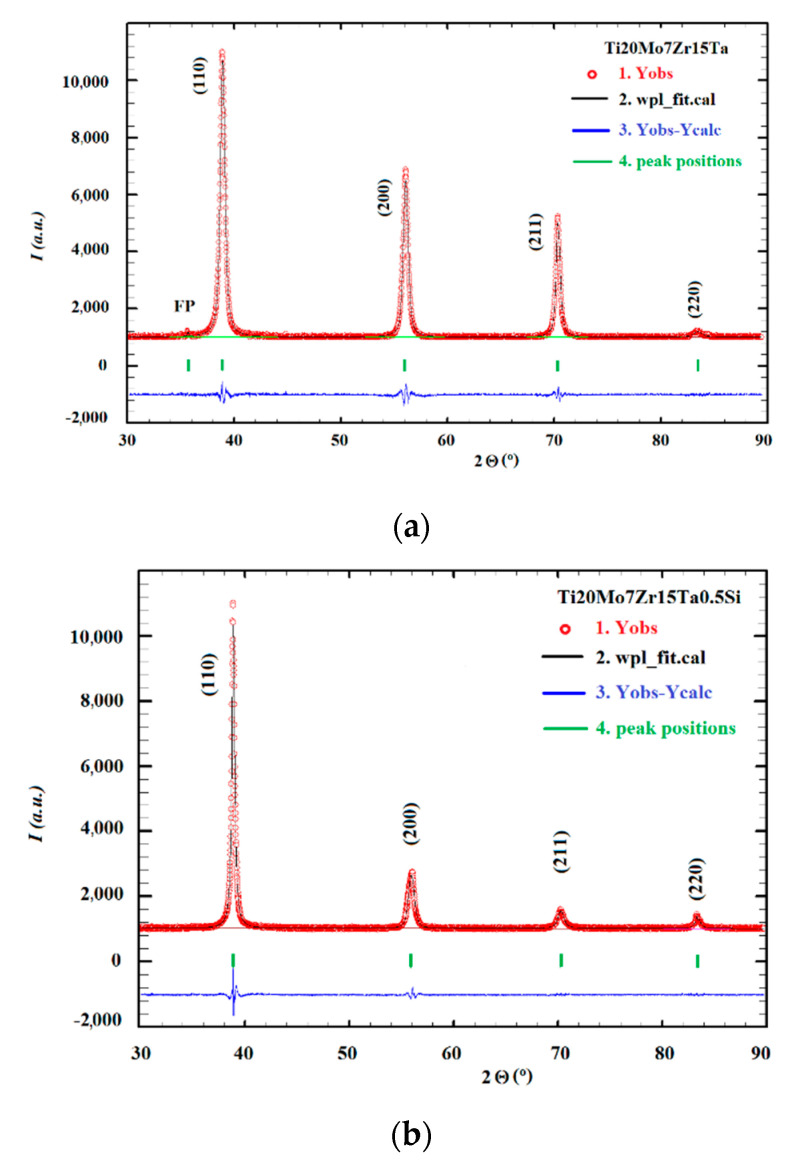
The observed (red), calculated (black) and the difference between the observed and calculated (blue) diffractograms of TiMo_20_Zr_7_Ta_15_Si_x_ corresponding to: (**a**) x = 0.0; (**b**) x = 0.5; (**c**) x = 0.75; (**d**) x = 1.0.

**Figure 4 materials-13-04808-f004:**
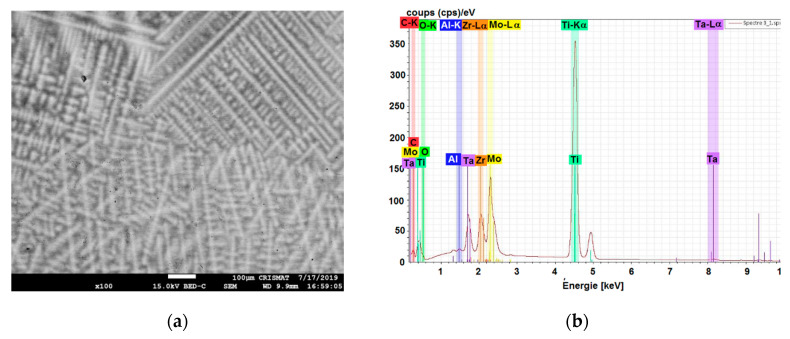
Microstructure and semi-quantitative composition spectrum for the sample #3: (**a**) SEM image; (**b**) EDS spectrum; (**c**–**f**) EDS maps of Ti, Ta, Mo, Zr—the scale is 200 µm.

**Figure 5 materials-13-04808-f005:**
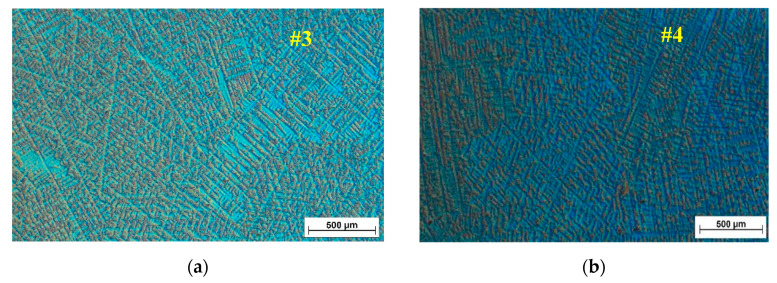
Optical microstructures for TiMo_20_Zr_7_Ta_15_Si_x_ samples with: (**a**) x = 0; (**b**) x = 0.5; (**c**) x = 0.75; (**d**) x = 1.0.

**Figure 6 materials-13-04808-f006:**
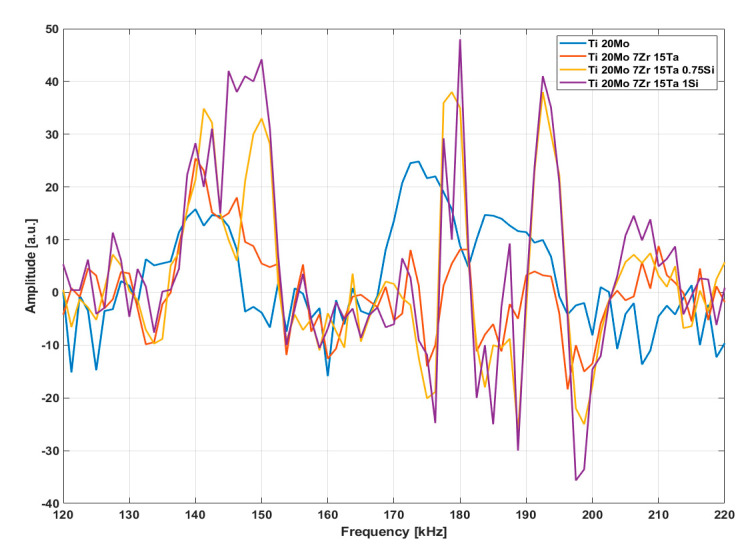
Resonance ultrasound spectra for samples shows the frequency shift, peaks splitting, and magnitude increasing.

**Figure 7 materials-13-04808-f007:**
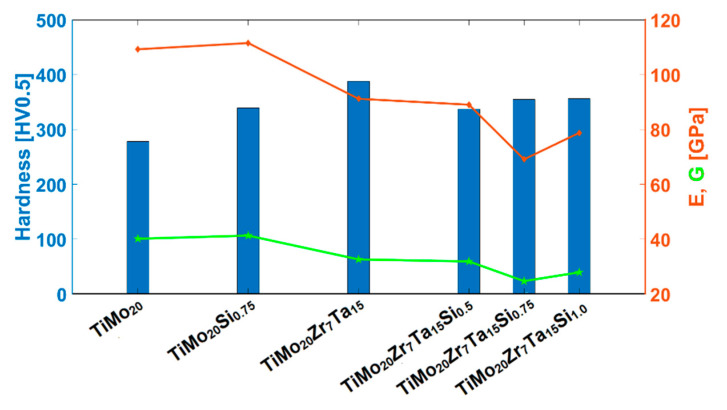
Mechanical properties of samples as a function of Si content.

**Figure 8 materials-13-04808-f008:**
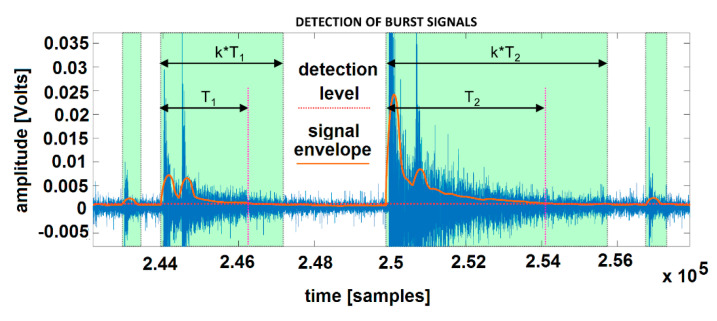
HD spectrogram computation.

**Figure 9 materials-13-04808-f009:**
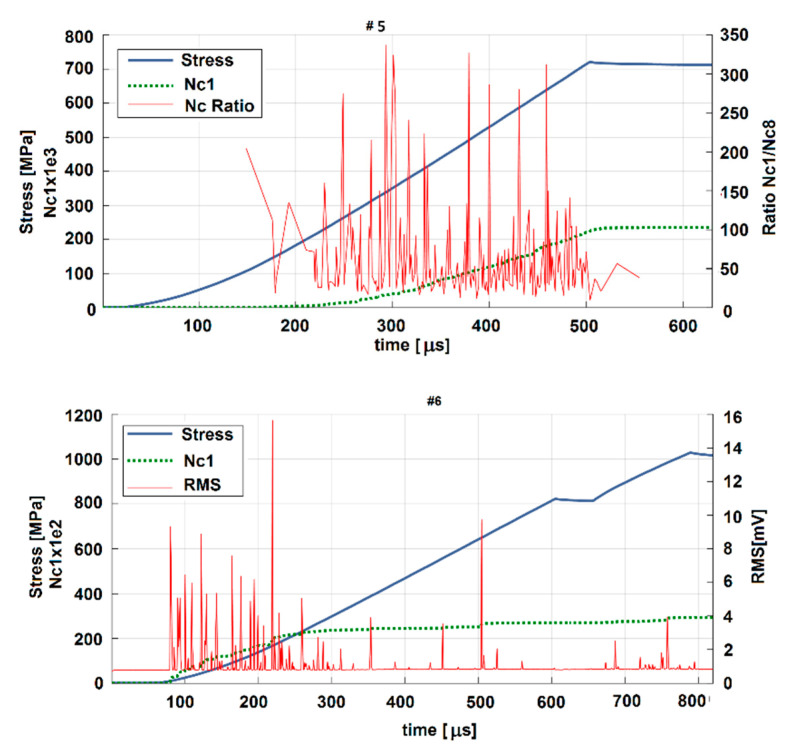
Acoustic emission results—spectral analysis.

**Table 1 materials-13-04808-t001:** Mechanical properties, dependent on Si content (x) for TiMo_20_Zr_7_Ta_15_Si_x_.

Sample	Composition	Density *ρ* [g/cm^3^]	Young Modulus E [GPa]	Shear Modulus G [GPa]	Poisson Ratio ν	C_l_ [m/s]	C_t_ [m/s]	HV0.5
#1	TiMo_20_	5.126 ± 0.004	109.23 ± 1.037	40.15 ± 0.329	0.36	5993 ± 0.7	2799 ± 0.5	278 ± 1.2
#2	TiMo_20_Si_0.75_	5.078 ± 0.006	111.51 ± 0.898	41.30 ± 0.304	0.35	5993 ± 0.5	2852 ± 0.4	339 ± 0.8
#3	TiMo_20_Zr_7_Ta_15_	7.049 ± 0.009	91.18 ± 0.248	32.61 ± 0.692	0.39	5222 ± 0.4	2151 ± 0.5	388 ± 1.7
#4	TiMo_20_Zr_7_Ta_15_Si_0.5_	6.845 ± 0.009	89.03 ± 0.456	31.87 ± 0.640	0.39	5210 ± 0.4	2158 ± 0.3	337 ± 1.1
#5	*TiMo_20_Zr_7_Ta_15_Si_0.75_	5.404 ± 0.001	69.11 ± 0.478	24.70 ± 0.260	0.39	5215 ± 0.4	2138 ± 0.4	354.8 ± 1.9
#6	TiMo_20_Zr_7_Ta_15_Si_1.0_	6.868 ± 0.005	78.78 ± 0.297	27.91 ± 0.276	0.41	5191 ± 0.5	2016 ± 0.4	356 ± 0.9

OBS * it is possible that after increasing the Si content over 0.75 wt.%, a supersaturation and distortions of the lattice appeared in the matrix, modifying the distance between atoms, and leading to modification of the elastic modulus.

**Table 2 materials-13-04808-t002:** Dependence of crystallographic structure on the Si concentration of samples of TiMo_20_Zr_7_Ta_15_Si_x_.

Sample	(x) Mass Concentration	a = b= c * (Å)	V (Å3)	D (Å) **	[ε ***]	B_o_	M_d_ (eV)	*ρ* *** (g/cm^3^)
#3	0.0	3.2785	35.239	204	0.001427	2.918	2.396	5.964
#4	0.50	3.2804	35.300	339	0.000844	2.917	2.395	5.931
#5	0.75	3.2818	35.346	790	0.000630	2.917	2.395	5.894
#6	1.00	3.2663	34.847	437	0.000967	2.916	2.394	5.952

* Lattice constants observed and determined using PowderCell. ** The average crystalline length and size of the microstrains were determined by means of the PowderCell program. *** Densities were calculated using the nominal compositions and experimentally determined constant lattices (PowderCell).

**Table 3 materials-13-04808-t003:** Atomic concentration of elements $\left(C_{i}^{T}{}^{i},C_{i}^{M}{}^{o},C_{i}^{Z}{}^{r},C_{i}^{T}{}^{a},C_{i}^{S}{}^{i}\right)$, calculated lattice constants (a_calc_ *) and calculated densities (*ρ* **).

Sample	$C_{i}^{Ti}(\%)$	$C_{i}^{Mo}(\%)$	$C_{i}^{Zr}(\%)$	$C_{i}^{Ta}(\%)$	$C_{i}^{Si}(\%)$	a_calc_ * (Å)	*ρ* ** (g/cm^3^)
#3	76.70	13.20	4.90	5.20	0	3.2714	6.003
#4	75.69	13.14	4.84	5.22	1.12	3.26620	6.008
#5	74.68	13.07	4.81	5.20	2.23	3.2551	6.041
#6	73.69	13.01	4.79	5.18	3.33	3.2477	6.055

* Lattice constants calculated by using the atomic radii of Ti, Si, Mo, Zr and Ta. ** Calculated densities with nominal compositions and calculated lattice constants (PowderCell).

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
