# Peer review of "Microstructural Analysis and Mechanical Properties of TiMo20Zr7Ta15Six Alloys as Biomaterials"

_materials, 2020, doi:10.3390/ma13214808_

Round 1

Reviewer 1 Report

In this manuscript authors carried out analysis and mechanical properties testings on Ti-Mo-Zr-Ta-Si alloys with varying composition in terms of Si wt%. Experiments are thorough and expertise in metallurgical knowledge are evident with this paper but there are few concerns;

1) The main variations of test groups are TiMo20SixZr7Ta15 alloys with varying Si concentration. Hence, it seems authors wanted to know what would happend when Si wt% is varied in this alloy. However, title does not reflect this. Also abtract seems occupied with background information that may not be relevant to the main focus of this study. Please reconsider title and abtract.

2) Introduction starts with stating "medical implants, in most specialities, are regulated by the FDA as class II device after demonstrating reasonable assurance of safety and efficacy." This statment is partly true. Implantable device as defintion would be class III in FDA which would be Class IIb or III in CE. Please revisit this information.

3) Although authors attempted to explain study carried out by previous paper (14 and 15), it was unclear how authors decided final wt% of Mo, Zr and Ta but only decided to vary Si. Clarifications woud be hepful to readers.

4) Authors stated importance of biological response of biomaterals but there are no information on such consideration for new alloy and varying Si. It would be useful to consider them.

Reviewer 2 Report

The paper is focues on the structural and mechanical properties of Ti-Mo-Zr-Ta-Si alloys. The topic falls within the scope of the journal. I recommend its publication after the following minor revisions:

  • Experimental details for the analysis of the microstructure should be added. For example, energy of beam and working distance should be reported for SEM analyses.
  • Table 1. I suggest to report the errors for all the mechanical parameters.
  • Figure 4. The scale length within the images is not clear. Please check and revise.

Reviewer 3 Report

The manuscript under consideration presents an extensive and well-presented study of mechanical properties of of the TiMoZrTa 393 quaternary alloy in the presence of doping with a variable percentage of Si. Evolution of mechanical characteristics was related to the modification of crystallographic structure. While the characterization techniques and results are reported adequately, the conclusion has to be made clearer and more understandable for the broad readership of the journal that may be interested in use of the studied material for, say, biological applications. Thus the conclusion section should answer the most important question: does Si doping have any sense and type of applications of the studied would benefit from this procedure. Same question should be addressed in the abstract section as well.  

When mentioning the resonance line splitting (line 408) authors should explain that they discuss the ultrasonic transmission.

X-axes Figure 8 should be label and calibrated based on the Si-concentration (x), not the sample number.

Reviewer 4 Report

A. Savin et al. present a systematic investigation of Ti-Mo-Zr-Ta-Si alloys. Overall, the language skills are below average. Unfortunately, some paragraphs are barely readable.

2.1 sample preparation: The authors study the impact of small variations in Si content. Have the exact alloy compositions been checked (ICP-OES)? Please add the results of the analyses.

L132: electron microscopy
L133: Brücker?
L215-217: move to "Materials and Methods"

Tab. 1 and 2: The calculation of the densities is not correct! I think the authors simply used the formula d = xa*da + xb*db + xc*dc... with x being the weight fraction. Mind the stoichiometry, i.e. the atom fractions must be inserted. Otherwise this formula must be used: 1/d = xa/da + xb/db + xc/dc... (x = weight fraction).
Thus, the alloy densities rather range between 5,83 and 5,90 kg/m3.

equation 2: Again, the authors inserted the weight fractions instead of the atom fractions. This is not correct!

Accordingly, all discussions involving densities and lattice parameters must be revised!

Figure 3: The resolution of the diffractograms is too low. Apparently, the blue line represents the difference between the diffractograms, not the green line. Fig (c): there is a texture effect, strong (110) and no (200) reflection? Any explanation?

Fig 4b: resolution too low

L266-269: In conclusion, there is only a weak influence of Si addition?

Fig 5: AFAIK, such a microstructure only forms upon deformation? Please explain the difference (treatment, preparation?) between fig. 5 and figs. 4&6. Corresponding images of x = 0 and 1 are missing. Furthermore, a proper analysis of the grain size (distribution) is required to support the statements.

Fig. 6: images a-c were taken with an optical filter (DIC?) whereas d) is apparently a bright field image. A valid comparison is only feasible when using the same preparation and imaging methods. L282-284: Please provide quantitative data supporting your statement that the distance between the dendrite axis first decreases and then increases again. Thus, Si content has a pronounced impact on the microstructure? Compare L266-269.

Fig. 8: all mechanical properties are already compiled in tab. 1. So, what is the purpose of this figure? Given the error bars of the hardness tests (#5-6), the minimum of the Young's modulus at 0.75 w% Si may not be very convincing.

The methods RUS and AE are completely unknown to me. I can only check the plausibility.

Fig. 10: too small, low resolution

L390-390: "The significantly higher AE signal in the case of alloys for which x = 1.0 compared to that for x = 0.75 can be explained by the differences between the microstructures." But the microstructures (fig. 6) and
the mechanical properties (fig. 8) are very similar.

Round 2

Reviewer 1 Report

All of my comments are well addressed with modifications in title, abstract and main texts.

Author Response

Response to Reviewer 1 Comments

All of my comments are well addressed with modifications in title, abstract and main texts.

Thank you for all support and for all your valuable time spent for revisions.

Reviewer 4 Report

Suggestions:

L31-35: Sentence too long. "...as(?) along..."
L132: Brüker --> Bruker
Table 2&3: Use sample index according to table 1, i.e. #3-6.
L243: [37]. and

Author Response

Response to Reviewer 4 Comments

Thank you for your careful revision and for your valuable time.

The changes are tracked in word manuscript.

L31-35: Sentence too long. "...as(?) along..."
Response 1. I have split the sentence.

L132: Brüker --> Bruker

Response 2: I made the correction

Table 2&3: Use sample index according to table 1, i.e. #3-6.

Response 3: I changed samples indexes in  the tables 2 and 3

L243: [37]. And

 Response 4: I deleted the point mark.
